# Interaction of Surface Water and Groundwater Influenced by Groundwater Over-Extraction, Waste Water Discharge and Water Transfer in Xiong'an New Area, China

**Meijia Zhu [1,2], Shiqin Wang [1,*], Xiaole Kong [1,2], Wenbo Zheng [1], Wenzhao Feng [1,2,3], Xianfu Zhang [4], Ruiqiang Yuan [5], Xianfang Song [6] and Matthias Sprenger [7,8]** 

1   Key Laboratory of Agricultural Water Resources, Center for Agricultural Resources Research, Institute of Genetics and Developmental Biology, Chinese Academy of Sciences, Shijiazhuang 050021, China; zhumeijia1994@163.com (M.Z.); xlkong@sjziam.ac.cn (X.K.); wbzheng@sjziam.ac.cn (W.Z.); pingyinghao1@sina.com (W.F.)
2   University of Chinese Academy of Sciences, Beijing 100049, China
3   School of Land Resources and Urban & Rural Planning, Hebei GEO University, No. 136 East Huai'an Road, Yuhua District, Shijiazhuang 050031, China
4   School of Water Resources and Environment, Hebei GEO University, No. 136 East Huai'an Road, Yuhua District, Shijiazhuang 050031, China; zhangxianfu@hgu.edu.cn
5   School of Environmental Sciences & Resources, Shanxi University, Taiyuan 030006, China; rqyuan@sxu.edu.cn
6   Key Laboratory of Water Cycle and Related Land Surface Processes, Institute of Geographic Sciences and Natural Resources Research, Chinese Academy of Sciences, Beijing 100101, China; songxf@igsnrr.ac.cn
7   Institute of Environmental Assessment and Water Research (IDAEA-CSIC), Barcelona 08034, Spain; mspreng@ncsu.edu
8   Department of Forestry and Environmental Resources, North Carolina State University, Raleigh, NC 27695, USA
*   Correspondence: sqwang@sjziam.ac.cn; Tel.: +86-0311-85810852; Fax: +86-0311-85815093

**Abstract:** Understanding the interaction of surface water and groundwater affected by anthropogenic activities is of great importance for water resource and water quality management. The Xiong'an New Area, located in the North China Plain, has been designated a new building area by China's government. Groundwater has been over pumped and artificial water was transferred to meet the water supply in this region. Therefore, the natural interaction of surface water and groundwater has been greatly changed and there has been a complex impact of the groundwater from anthropogenic activities. In this study, we used water chemical ions and stable isotopes of $\delta^2$H and $\delta^{18}$O to assess the interaction of surface water and groundwater in the Xiong'an New Area. We carried out field surveys and water sampling of the Fu River (domestic waste water discharge), Lake Baiyangdian (artificial water transfer), and the underlying groundwater along the water bodies. Results show that the artificial surface water (discharged and transferred) became the major recharge source for the local groundwater due to the decline of groundwater table. We used groundwater table observations, end-member mixing analysis of the stable isotopic composition and chloride tracers to estimate the contributions of different recharge sources to the local groundwater. Due to the over pumping of groundwater, the lateral groundwater recharge was dominant with a contribution ratio ranging from 12% to 78% in the upper reach of the river (Sections 1–3). However, the contribution of lateral groundwater recharge was estimated to be negligible with respect to the artificial water recharge from Lake Baiyangdian. Seepage from the Fu River contributed a significant amount of water to the connecting aquifer, with a contribution ranging from 14% to 75% along the river. The extent of the river influence into the aquifer ranges as far as 1400 m to the south and 400 m to the north of the

Fu River. Estimations based on isotopic fractionation shows that about 22.3% of Lake Baiyangdian water was lost by evaporation. By using the stable isotopes of oxygen and hydrogen in the lake water, an influencing range of 16 km west of the lake was determined. The interaction of the surface water and groundwater is completely changed by anthropogenic activities, such as groundwater over pumping, waste water discharge and water transfer. The switched interaction of surface water and groundwater has a significant implication on water resources management.

**Keywords:** surface water—groundwater interaction; anthropogenic activities; groundwater over-exploitation; groundwater recharge; end-member mixing analysis (EMMA); North China Plain; Xiong'an New Area

---

## 1. Introduction

Surface water and groundwater are an interrelated hydrological continuum, which must be considered in the calculation of hydrological cycle and water budget [1,2]. Understanding pathways and quantifying the fluxes between surface water and groundwater systems are essential to evaluate water resource allocations and to assess potential impacts of increasing water use on groundwater-dependent ecosystems [2–4].

Anthropogenic activities, such as groundwater over pumping and artificial water discharge or transfer, are among the main factors influencing the interaction between surface water and groundwater [5,6]. These regional water balance changes can reduce the hydraulic connectivity and result in pollution of water resources [2]. On the one hand, excessive extraction of groundwater interferes with surface water and groundwater systems, resulting in reduction of stream flow and ecological degradation [7,8]. Because of the extensive agricultural irrigation, groundwater exploitation is relatively intensive, and the spatial and temporal distribution of the groundwater level has changed dramatically, especially in arid and semi-arid areas [9]. The decline of the groundwater level leads to a decreased runoff and potential dry out of rivers. On the other hand, artificial water transferred into rivers and lakes and waste water from urban areas discharged into rivers became new recharge sources for groundwater. Particularly, sewage produced by local human activities led to the pollution of rivers [10,11]. With industrialization and accelerated urbanization, the leakage of sewage has adverse effects on groundwater quality [12–14]. Since groundwater and surface water may be polluted by different sources and kinds of pollutant species, quantifying the amount of induced infiltration is an important factor to evaluate groundwater quality. However, the variation of interaction of surface water and groundwater is dramatic in regions with groundwater over-extraction, waste water discharge and water transfer. Therefore, understanding the interactions of the surface water and groundwater under the influence of anthropogenic activities is important for water resources management and water pollution prevention and treatment [15].

Hydrochemistry and environmental isotopic techniques have frequently been employed to study the interaction of surface water and groundwater [15–20]. As components of water molecules, the stable isotopes $^{18}O$ and $^2H$ are ideal natural tracers for studying the water cycle [21,22]. By using the tracing principle of stable isotopes, the transformation relationship between surface water and groundwater in different areas can be clarified and the amount of exchange between them can be calculated [23]. Chloride ($Cl^-$) is a conservative tracer, which was used to estimate recharge sources [24–27]. End-member mixing analysis can be used to evaluate the contribution ratio of each recharge source of groundwater [19,28,29].

The water shortage of the North China Plain (NCP) is serious, where groundwater provides more than 70% of the total water supply for agricultural irrigation [30,31]. Reservoirs built in the mountain area and over exploitation of groundwater in the plain area resulted in the reduction of river waters, a decline of the groundwater level, and enlargement of the groundwater cone

area [32]. Lake Baiyangdian (BYD) is the largest freshwater lake in the NCP. Historically, the lake water was recharged by inflowing river water and groundwater. Extensive reservoir constructions in the upstream regions of the lake catchment have further worsened the lake hydrology by cutting off river contributions into the lake [33,34]. Previous studies showed that the climate variations accounted for 38–40% of stream flow decrease, while human activities accounted for 60–62% in this lake catchment [35]. Intensive groundwater extraction for agricultural irrigation is another factor for the drastic groundwater drawdown in the lake catchment and the wider NCP [36,37]. The decline of the groundwater level resulted in surface water being the direct recharge source for groundwater [38]. To resolve the water shortage, water was transferred from upstream reservoirs or the Yellow River to satisfy the water demand of the lake. Beside the freshwater transferred into the lake, waste water from the urban areas is discharged into the river and flows into the lake [39]. In April 2017, Chinese government declared to establish a new area near Lake BYD which was named the Xiong'an New Area. More water will be transferred to satisfy the water needs of the Xiong'an New Area and more waste water will be discharged to the river. This will disturb greatly the interaction of river, lake and groundwater. Therefore, our objective is to understand the interaction of surface water and groundwater and its influencing factors on this river–lake groundwater system in the Xiong'an New Area. This will benefit water management and water transfer planning during the development of the area in the near future.

In this paper, a river–lake groundwater system in the Xiong'an New Area, the Lake BYD watershed of the NCP, was selected as the case study area to study the interaction of surface water and groundwater by using the multiple tracer methods. The aims of this study include (1) identifying the recharge sources of groundwater and its influencing factors; and (2) characterizing the impact of groundwater over-extraction, waste water discharge and water transfer on surface water-groundwater interaction.

## 2. Study Area

### 2.1. Site Description

Lake BYD is the largest freshwater lake in the NCP, located in the core area of Xiong'an New Area, downstream of the Lake BYD watershed. Baoding city is located in the upstream of the Fu River which flows into the Lake BYD (Figure 1). The region is characterized by a temperate continental monsoon climate with an annual average rainfall of 510 mm/year [40] and evaporation of 1369 mm/year [41]. The majority of the precipitation (75%) falls from June to September (Figure 2). The mean annual air temperature is 13.8 °C and the mean absolute humidity is 61% based on data obtained from the Baoding meteorological station of the China meteorological data-sharing service [14,42].

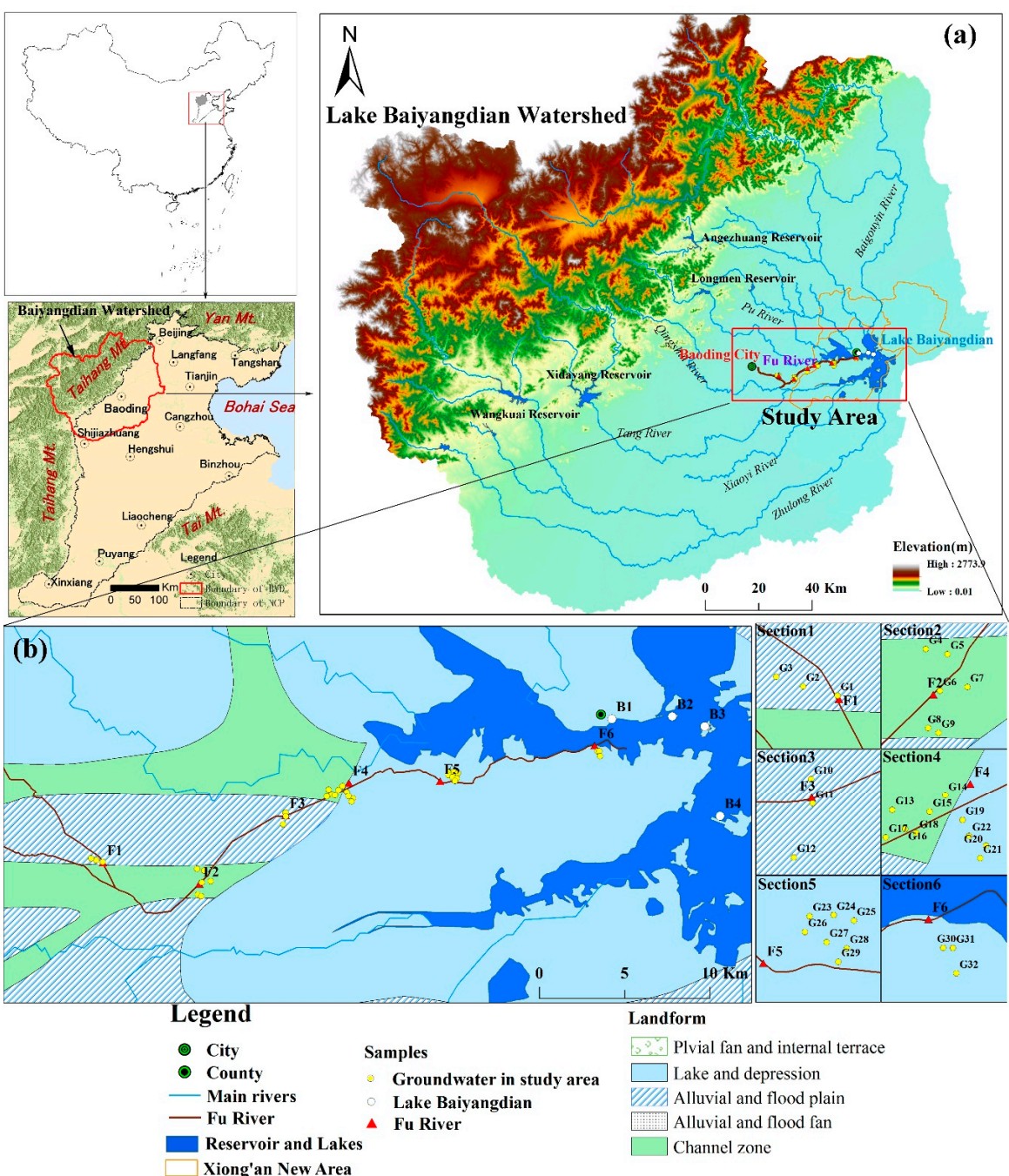

**Figure 1.** Location of the study area in Lake Baiyangdian (BYD) watershed (**a**) and sampling sites in six sections from the Fu River to the Lake BYD watershed (**b**).

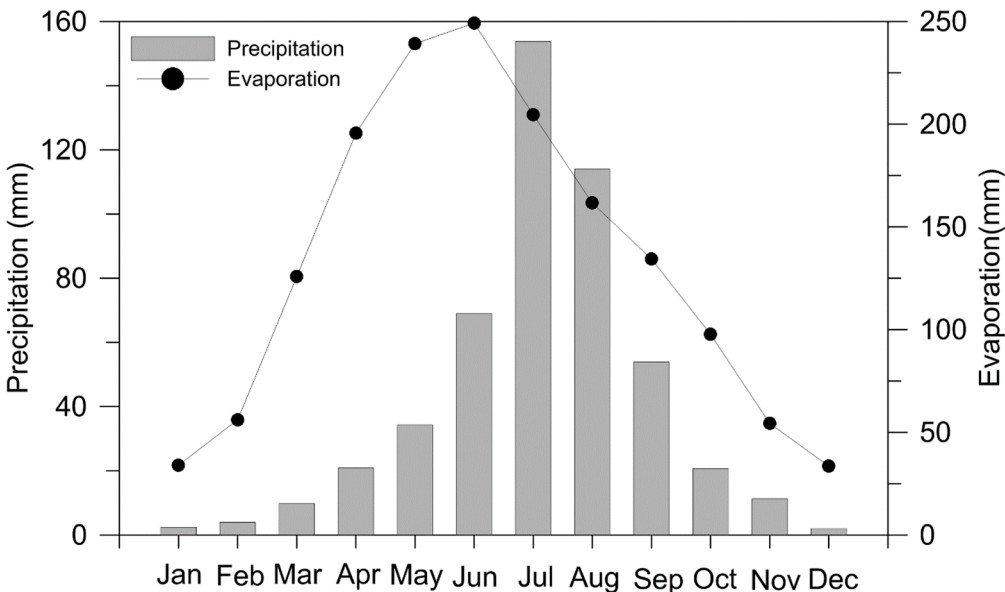

**Figure 2.** Mean monthly precipitation and evaporation in Baoding City from 1981 to 2010.

## 2.2. Hydrogeology

The geology of the study area is composed of unconsolidated sediments of Quaternary. Silt clay and clay are widely distributed in the aquifer [14] (Figure 3). The aquifer can be divided into four groups according to its stratigraphic features (aquifer groups I, II, III and IV) [43]. The first and second aquifer groups (I, II) include aquifers of Holocene $Q_h$ and the upper Pleistocene $Q_p^3$. The third aquifer group (III) is the middle Pleistocene $Q_p^2$. The fourth aquifer group (IV) is the Lower Pleistocene $Q_p^1$ aquifer group [36]. Groundwater is divided into shallow groundwater with a depth less than 100 or 120 m and deep groundwater with depth greater than 100 or 120 m [14]. Previous research showed that there is weak hydraulic connection between shallow and deep groundwater [14,36,44]. Therefore, only the shallow groundwater was considered in this study.

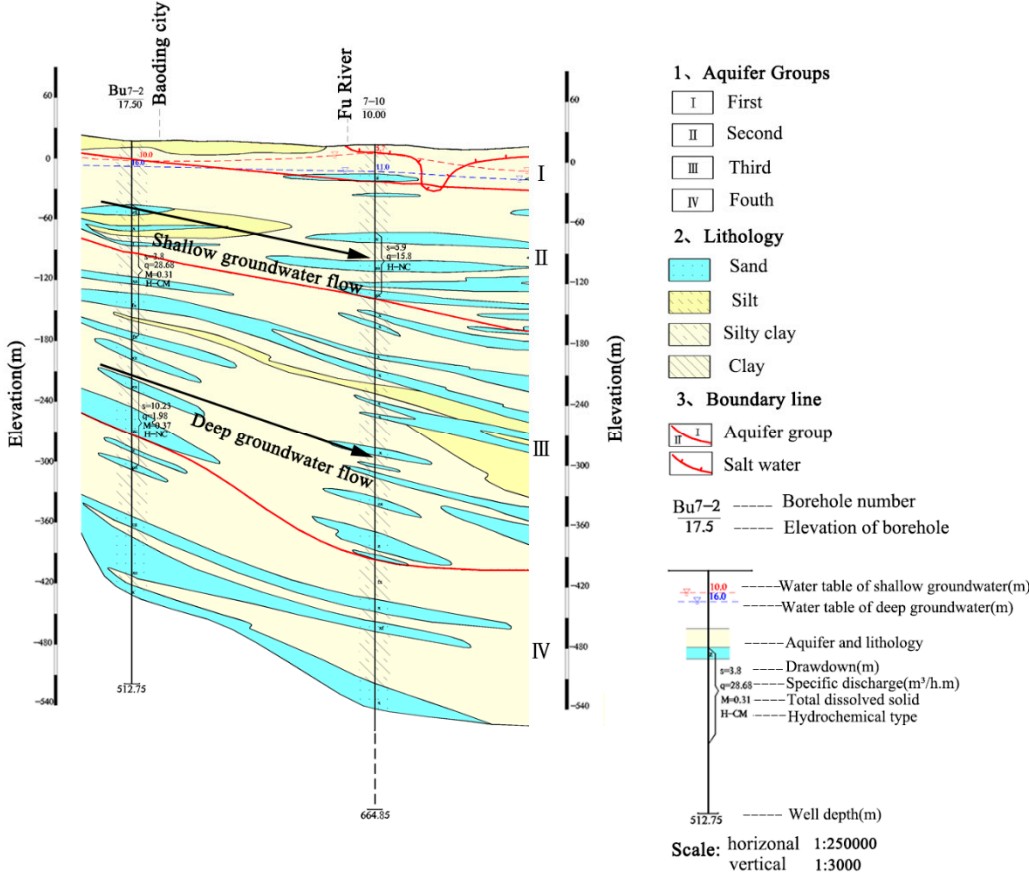

**Figure 3.** Hydrogeological profile from Baoding City to Fu River (modified according to Zhang et al. [45]).

*2.3. Hydrological Setting*

Historically, eight rivers were flowing into the lake: Baigouyin River, Pu River, Cao River, Fu River, Tang River, Xiaoyi River, Zhulong River and Ping River [36]. At present, the Fu River, which is mainly composed of sewage and industrial waste water from Baoding city, is the only river that has perennial flow into the Lake BYD [46]. The discharge of waste water in Baoding City is 273.79 million t/a, of which 213.09 million tons of industrial waste water accounts for 77.8% of the total waste water. About $1 \times 10^5$ m$^3$ of domestic sewage and waste water flows into the river per day, accounting for 45.2% of the average flow rate of the river [46]. Thus, the Fu River with domestic waste water is a great threat to the water quality of Lake BYD and the groundwater connected to it.

Beside the inflow of sewage, water transfer measures were carried out many times to the Lake BYD. Water was mainly transferred from the upstream Angezhuang Reservoir, Wangkuai Reservoir, Xidayang Reservoir, and Yuecheng Reservoir and the Yellow River across the basin [47]. On 37 occasions water was transferred between 1981 to 2018, of which 29 were transfers from upstream reservoirs, with an amount of $1.67 \times 10^9$ m$^3$ (Table S1). Figure 4 shows the quantity of water transferred and flowed into the lake from 2009 to 2018. Water loss along the water transfer line accounted for 27 to 73% of the total transferred water. The water quantity transferred in 2017 and 2018 was largest because the establishment of the Xiong'an New Area was declared.

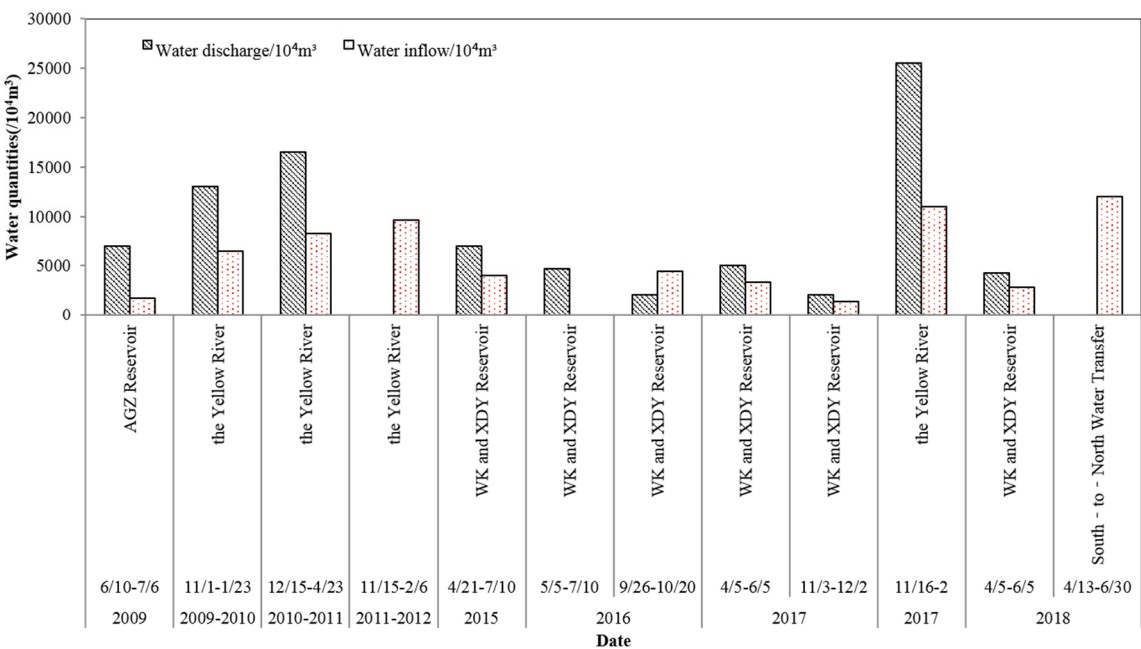

**Figure 4.** Statistics of water transfer for Lake BYD from 2009 to 2018. AGZ Reservoir = Angezhuang Reservoir; WK Reservoir = Wangkuai Reservoir; XDY Reservoir = Xidayang Reservoir.

## 3. Materials and Methods

### 3.1. Water Sampling and Laboratory Analyses

We sampled water from the Fu River, Lake BYD and the groundwater along the Fu River (Figure 1b). Sampling periods covered the dry season (January and March 2018) and rainy season (June and August 2018). There were six river water sampling locations upstream of Lake BYD (F1–F6). Perpendicular to the river channel, we sampled at 32 groundwater wells in six sections on both sides of the Fu River with different distances to the river (G1–G32). There were four sampling locations in the lake (B1–B4). We further sampled the Lake BYD water in September 2008, June 2009, May 2010, July 2014, June 2016, December 2016, and May 2018.

The groundwater depth was measured at each well in situ using a water level gauge (Model102, Solinst, Georgetown, ON, Canada) before each water sampling. In parallel, temperature (*T*), electrical conductivity (EC), and pH were measured in situ using a portable meter (WM-22EP) (DKK, TOA Corporation, Tokyo, Japan). Before sampling, the groundwater was pumped for about 3 min to remove stagnant well water. All water samples were filtered through a 0.22 μm filter membrane before using ion chromatograph (ICS-600, Dionex, Sunnyvale, CA, USA) to determine $Na^+$, $K^+$, $Mg^{2+}$, $Ca^{2+}$, $Cl^-$, $NO_3^-$, $SO_4^{2-}$ concentrations. Bicarbonate concentrations were measured by titration with normal diluted 0.01 N $H_2SO_4$ immediately after samples were taken back to the laboratory. Stable isotopes $^2H$ (±0.5‰) and $^{18}O$ (±0.2‰) of water were measured by liquid water isotope analyzer (Picarro-i2120, Picarro, Santa Clara, CA, USA). The isotope ratios were expressed in the standard $\delta$-notation as per mil (‰) difference from standard-VSMOW (Vienna Standard Mean Ocean Water) [48]. All analyses were conducted at the Centre for Agricultural Resources Research, Chinese Academy of Science (CAS).

### 3.2. Rayleigh Evaporation Model

When water is transferred into the Lake BYD, evaporation is the major water loss resulting in isotopic enrichment. The evaporation processes for the lake satisfies the condition of Rayleigh-type fractionation [49], which has been used by Wang et al. [14]. A simple model of $\delta^2H$ and $\delta^{18}O$ can be developed incorporating both the equilibrium and kinetic enrichment factors of water that has undergone evaporation [50]. The general form of a Rayleigh fractionation equation states that the

isotope ratio of the reactant in a diminishing water pool is a function of its initial isotopic ratio ($R_0$), the remaining fraction of the water pool ($f$) and the fractionation factor ($\varepsilon$, $\varepsilon = \alpha_{v-w} - 1$) for the reaction, which incorporates both equilibrium ($\varepsilon_{w-v}$) and kinetic fractionation ($\Delta\varepsilon_{v-b1}$) [51].

$$R = R_0 f^{(\alpha_{v-w}-1)} \tag{1}$$

After converting the isotope ratios to $\delta$ values, Equation (1) can be given as:

$$\delta = \exp[\ln(f) \times (\varepsilon)/1000] \times (\delta_0 + 1000) - 1000 \tag{2}$$

The equilibrium isotope fractionation factor is dependent on the temperature ($T$, in Kelvin degree) [52,53]. The following equations were adopted to calculate the equilibrium fractionation factors of $^{18}O$ and $^2H$ between water and vapor (expressed as $10^3 \ln \alpha^{18}O_{v-w}$ and $10^3 \ln \alpha^2 H_{v-w}$) [52].

$$10^3 \ln \alpha^{18}O = 1.137\left(10^6/T^2\right) - 0.4156\left(10^3/T\right) - 2.0667 \tag{3}$$

$$10^3 \ln \alpha^2 H = 24.844\left(10^6/T^2\right) - 76.248\left(10^3/T\right) + 52.612 \tag{4}$$

Gonfiantini described the kinetic effects in terms of humidity ($h$) using the following relationships [54]:

$$\Delta\varepsilon^{18}O = 14.2(h-1) \tag{5}$$

$$\Delta\varepsilon^2 H = 12.5(h-1) \tag{6}$$

where, $\Delta\varepsilon^{18}O_{v-b1}$ and $\Delta\varepsilon^2 H_{v-b1}$ are kinetic fractionation factors of $^{18}O$ and $^2H$, respectively, between vapor and boundary layer in the evaporation interface.

### 3.3. End-Member Mixing Analysis (EMMA)

Stable isotopes ($\delta^{18}O$) and chloride ($Cl^-$) were used as tracer information to estimate the contribution ratios of recharge sources for groundwater. If the three end-members are mixed, the $\delta$-C value (isotopic composition-trace ion concentration) of the mixture must fall within the area enclosed by the three end-members. The formula is as follows:

$$f_a + f_b + f_c = 1 \tag{7}$$

$$\delta_a f_a + \delta_b f_b + \delta_c f_c = \delta_m \tag{8}$$

$$C_a f_a + C_b f_b + C_c f_c = C_m \tag{9}$$

where, $\delta$ is the concentration of isotope tracer; $C$ is concentration of $Cl^-$; $a$, $b$ and $c$ represent three sources of water; $m$ is the resulting mixed water.

Due to the limitation of sampling dates and locations, uncertainty exists in the compositions/concentrations of tracers for end-members and resultant mixtures. Therefore, the standard errors of contribution ratios estimated by end-member mixing analysis (EMMA) were computed by error propagation analysis [55].

## 4. Results and Discussion

### 4.1. River Water Level and Groundwater Table

Figure 5 shows the river water level and groundwater table observed at each section. The river water level was higher than the groundwater table in Sections 1–5, indicating surface water recharging to groundwater. The decline of the groundwater table resulted in the surface water becoming the major recharge source for groundwater [56]. Additionally, the groundwater table in the north side of the river

was higher than that of the south side, consistent with the flow direction of the regional groundwater flow from northwest to southeast in the watershed [36]. However, the river water level at the outlet of the Fu River (Section 6) was close to or below the groundwater table. Groundwater depth also showed a decreasing trend from west (>10 m in Sections 1–3) to east (<5 m in Sections 4–6) (Figure 6a).

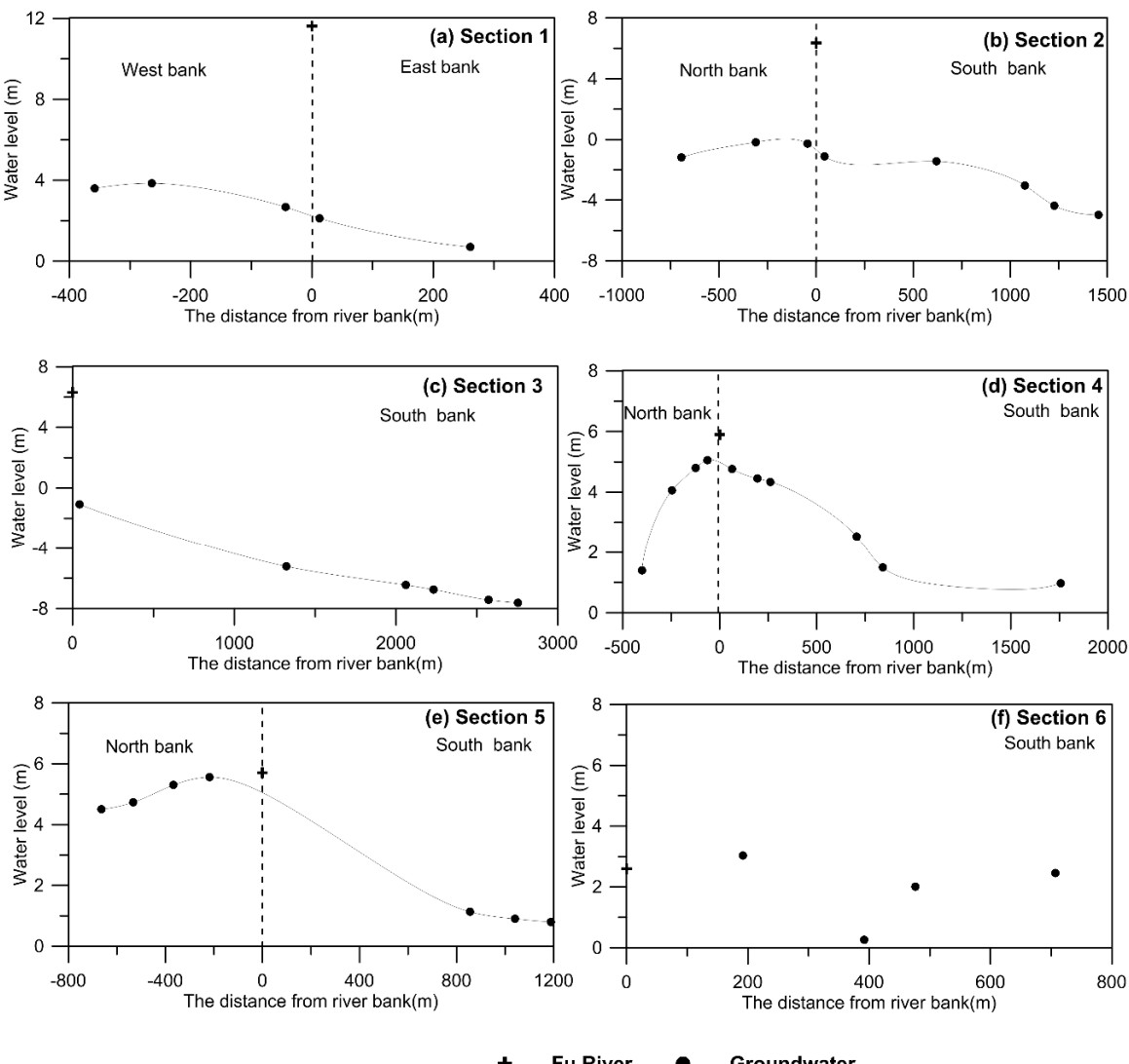

**Figure 5.** River water level and groundwater table in each section.

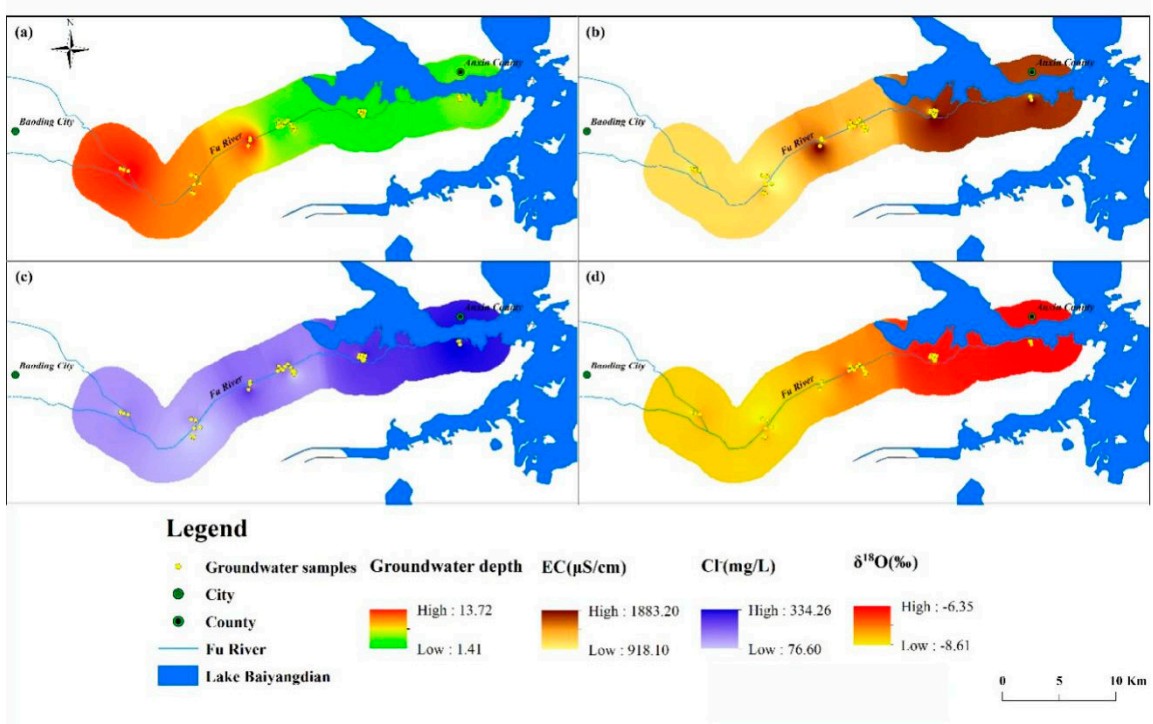

**Figure 6.** Spatial interpolation of groundwater depth (**a**), electrical conductivity (EC) (**b**), Cl$^-$ (**c**) and $\delta^{18}$O (**d**) of groundwater along the Fu River.

*4.2. Hydrochemical Characteristics*

Major ion composition and ionic ratios can act as a track-record of water-rock interaction during flow [57,58]. The statistical data of water chemicals are listed in Table 1. The mean pH for the Lake BYD water, Fu River water and groundwater were 8.1, 7.7 and 7.5, respectively. The lake water was composed of the transferred water from reservoirs upstream or the Yellow River water which are slightly alkaline. The relative low EC value of the lake water also indicated the water quality variation of transferred water with a range from 676 to 1297 μS/cm and a mean value of 967 μS/cm. Mean EC values of Fu River water (1286 μS/cm) and groundwater (1356 μS/cm) were similar to each other, suggesting a connection of the two water bodies. EC of Fu River water ranged from 1010 to 1682 μS/cm, with a low coefficient of variation (C.V.) of 14.5%, which results from the stable discharge of domestic sewage from Baoding city.

EC of the groundwater had the largest range from 747 to 2600 μS/cm with a C.V. value of 23.9%. This revealed that the influencing extent of surface water recharge on groundwater varied spatially as shown in Figure 6b. EC values were lower than 1000 μS/cm in Sections 1 and 2, ranged between 1500 and 1800 μS/cm in Section 3, ranged between 1200 and 1500 μS/cm in Section 4, and were higher than 1800 μS/cm in Sections 5 and 6.

Total dissolved solids (TDS) values of the Fu River water, Lake BYD water, and groundwater had similar trends as EC. Chloride is considered as one of the indicative ions from domestic sewage. The mean concentration sequence of Cl$^-$ in different water bodies was: Fu River water > groundwater > Lake water. The highest Cl$-$ concentration of the Fu River water was 426 mg/L. The Cl$-$ concentration also increased from west to east. They were lower than 150 mg/L in Section 1, 2 and 4 and above 250 mg/L in Sections 5 and 6 (Figure 6c).

**Table 1.** Statistical values of field investigation parameters and major ion mass concentrations.

| Type | Number | Item | pH | EC /μS·cm$^{-1}$ | Ca$^{2+}$ | Mg$^{2+}$ | Na$^+$ | K$^+$ | Cl$^-$ /mg·L$^{-1}$ | SO$_4$$^{2-}$ | NO$_3$$^-$ | HCO$_3$$^-$ | TDS /g·L$^{-1}$ |
|---|---|---|---|---|---|---|---|---|---|---|---|---|---|
| Fu River water | 23 | Min | 7.12 | 1010.7 | 84.8 | 33.4 | 118.4 | 17.7 | 151.7 | 6.4 | 26.2 | 300.6 | 0.70 |
| | | Max | 7.99 | 1682.0 | 110.8 | 43.2 | 261.7 | 32.1 | 425.7 | 119.0 | 74.5 | 506.9 | 1.15 |
| | | Mean | 7.67 | 1285.9 | 93.6 | 36.5 | 191.8 | 22.7 | 290.5 | 70.5 | 51.1 | 376.1 | 0.94 |
| | | C.V. (%) | 2.92 | 14.5 | 8.3 | 7.3 | 28.3 | 15.3 | 32.2 | 46.6 | 29.0 | 15.5 | 0.01 |
| Lake water | 28 | Min | 7.10 | 676.0 | 44.7 | 21.9 | 49.2 | 5.2 | 54.4 | 56.5 | 0.0 | 160.7 | 0.38 |
| | | Max | 8.92 | 1297.0 | 82.7 | 49.3 | 214.4 | 19.5 | 308.2 | 391.9 | 34.0 | 385.8 | 0.96 |
| | | Mean | 8.11 | 966.8 | 63.9 | 33.3 | 104.6 | 12.6 | 128.6 | 121.3 | 5.7 | 282.6 | 0.61 |
| | | C.V. (%) | 4.25 | 19.5 | 13.9 | 20.5 | 36.1 | 25.8 | 42.2 | 55.7 | 130.2 | 17.1 | 0.02 |
| Groundwater | 107 | Min | 6.43 | 746.8 | 29.3 | 33.0 | 41.4 | 0.0 | 58.8 | 65.3 | 0.0 | 324.7 | 0.53 |
| | | Max | 7.96 | 2600.3 | 242.2 | 249.4 | 296.5 | 19.5 | 339.4 | 464.6 | 291.4 | 1692.4 | 1.92 |
| | | Mean | 7.45 | 1356.2 | 109.6 | 106.7 | 136.4 | 3.6 | 186.9 | 159.0 | 39.7 | 584.5 | 1.03 |
| | | C.V. (%) | 3.31 | 23.9 | 44.4 | 42.7 | 41.0 | 84.5 | 37.7 | 48.9 | 121.6 | 31.0 | 0.03 |

Figure 7 shows the piper diagram for all samples in the study area. Water types of the Fu River water samples were mainly Na·Ca-Cl·HCO$_3$ and Ca·Na·Mg-HCO$_3$·Cl. While the hydrochemical types of the lake water mainly included Ca·Na·Mg-HCO$_3$·Cl, Na·Ca-Cl·HCO$_3$, Ca·Mg-Cl·HCO$_3$, Na·Ca·Mg-HCO$_3$·Cl, Na·Ca·Mg-HCO$_3$·Cl·SO$_4$ and Na·Mg·Ca-HCO$_3$·Cl·SO$_4$ in different sampling periods. The diversity of water types suggests the influence of different water sources of the lake including waste water of the Fu River water, the upstream reservoir water, and the transferred water by inter-basin water transfer project (Figure 4). The hydrochemical types of the lateral groundwater upstream of the watershed and precipitation were Ca·Mg-HCO$_3$, and Ca-HCO$_3$·SO$_4$, respectively [36]. Most of the groundwater near the Fu River plotted between the lateral groundwater and the surface water in Figure 7, suggesting mixing of these two water bodies. In particular, the hydrochemical types of groundwater in Sections 4–6 (Mg·Na·Ca-HCO$_3$·Cl, Na·Mg·Ca-HCO$_3$·Cl, Ca·Mg·Na-HCO$_3$·Cl) were similar to those of the Fu River water and the Lake BYD water, indicating the influence of surface water. In addition to the influence of the Fu River and the Lake BYD, it was further reported that fertilization, sewage irrigation and point source pollution also affected the groundwater quality in this region [36].

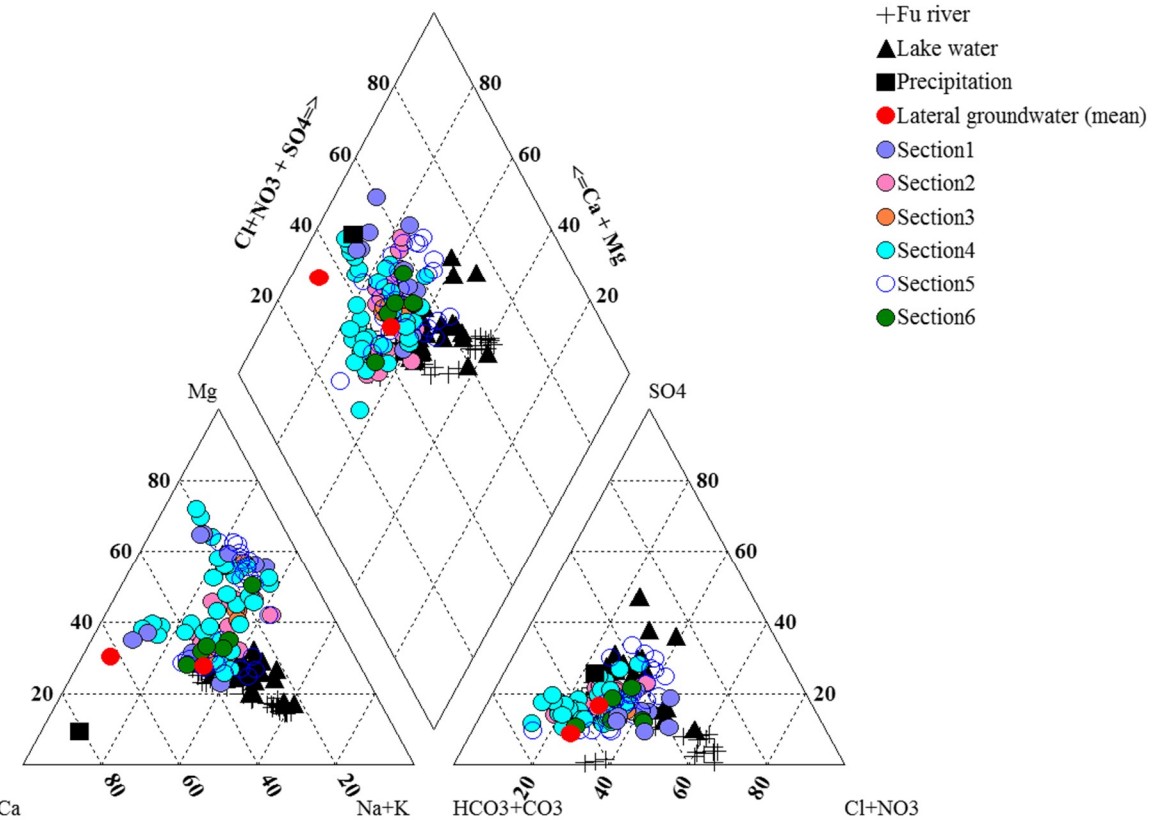

**Figure 7.** Piper diagram of surface water and groundwater in the study area. The hydrochemical characteristics of precipitation come from the mean ion mass concentration of precipitation in the Luancheng Agro-Ecosystem Experimental Station, Chinese Academy of Science (CAS).

*4.3. Isotopic Composition and Evaporation Estimates*

$\delta^2$H and $\delta^{18}$O values of groundwater are summarized in Table 2. The lake water was most enriched in stable isotopes with average $\delta^{18}$O and $\delta^2$H values of −5.7‰ and −47‰, respectively; ranging from −1.4‰ to −7.7‰ and −57‰ to −28‰, respectively. The mean values for groundwater were −7.5‰ and −57‰ for $\delta^{18}$O and $\delta^2$H, respectively; varying from −8.7‰ to −6.0‰ and −64‰ to −49‰, respectively. The isotopic compositions of the Fu River water for ranged from −8.4‰ to −7.3‰ for $\delta^{18}$O and from −61‰ to −54‰ for $\delta^2$H with a mean value of −8.0‰ for $\delta^{18}$O and −58‰ for $\delta^2$H. The spatial variation of the isotopic compositions of the Fu River water was smaller than for the

groundwater and lake water. The water samples were enriched in $\delta^{18}O$ from west to east (Figure 6d) with values below $-8‰$ in Sections 1–3 and above $-6‰$ in Sections 4–6.

**Table 2.** Statistical values of $\delta^2H$ and $\delta^{18}O$ in surface water and groundwater.

| Type | $\delta^{18}O$ (‰) | | | | $\delta^2H$ (‰) | | | |
|---|---|---|---|---|---|---|---|---|
| | Min | Max | Mean | C.V. (%) | Min | Max | Mean | C.V. (%) |
| Fu River water | −8.4 | −7.3 | −8.0 | −3.9 | −61 | −54 | −58 | −3 |
| Lake water | −7.7 | −1.4 | −5.7 | −28.3 | −57 | −28 | −47 | −17 |
| Groundwater | −8.7 | −6.0 | −7.5 | −8.4 | −64 | −49 | −57 | −6 |

The relationship between $\delta^2H$ and $\delta^{18}O$ for all water samples is shown in Figure 8. All of the samples plot below the local meteoric water line (LMWL), indicating that all surface water and groundwater experienced evaporation. The $\delta^2H$ and $\delta^{18}O$ values of lake water were larger than those of the Fu River water due to evaporation when river water flowed and recharged into the lake. In winter, the reservoir water depleted in heavy isotopes was transferred to the lake, resulting in a decline of the isotopic composition in the lake water (e.g., water transferred in January and March 2018 in Figure 8). No water was transferred to the lake before the rainy season and $\delta^{18}O$ and $\delta^2H$ values were highest (e.g., lake water in May 2010 and June 2009). An evaporation line was fitted for the lake water ($\delta^2H = 4.49\delta^{18}O - 22.07$, $R^2 = 0.98$). The low slope of 4.49 also shows the strong evaporation effect of the lake water. All groundwater samples were distributed around the evaporation line of the lake water, suggesting the impact of surface water on groundwater. Since the lake water was most enriched in isotopes, the leakage of the lake water led to increased isotopic compositions in groundwater near the lake than in groundwater further away from the lake. Therefore, the $\delta2H$ and $\delta18O$ compositions in groundwater showed an increasing trend from Sections 3–6 except for Sections 1 and 2 (Figure 8).

As the water transferred into Lake BYD was stored for maintaining wetland ecosystems and lost by evaporation and leakage to groundwater. The Rayleigh distillation of isotopes can be used to estimate the evaporation ratios. The application of this theory assumes the following conditions described by Wang et al. [14]: (1) the lake is a steady state system with fixed inflow and zero outflow; (2) the water is fully mixed; and (3) all water is available for evaporative enrichment. The average annual temperature of Baoding City (13.8 °C) was adopted in Equations (3) and (4) to calculate the equilibrium fractionation factors. Correspondingly, the vapor–water isotope enrichment factors were $-84.7‰$ and $-10.2‰$ for $\delta^2H$ and $\delta^{18}O$, respectively. The lake was mixed with different water sources. Therefore, the average isotope values of different water sources of the lake taken as the initial average value for the evaporation of the lake water were $-58.3‰$ and $-8.05‰$ for $\delta^2H$ and $\delta^{18}O$, respectively. The slope of the evaporation line of the lake water (4.49) was used to find the best-fit line when using varying humidity values. The best-fit line ($\delta^2H = 5.01\delta^{18}O - 18.24$) was obtained when the humidity value was 50%, which is close to the average minimum humidity of 53% during dry seasons [14]. The evaporation ratio of the lake was calculated according to the evaporation rate in different seasons, with September 2008, June 2009, May 2010, July 2014, respectively. The results showed that the average evaporation loss of Lake BYD accounted for 22.3% of the total water input. Correspondingly, 77.7% of the lake water was lost by leakage and agricultural irrigation return recharge to groundwater without considering the direct usage of the lake water.

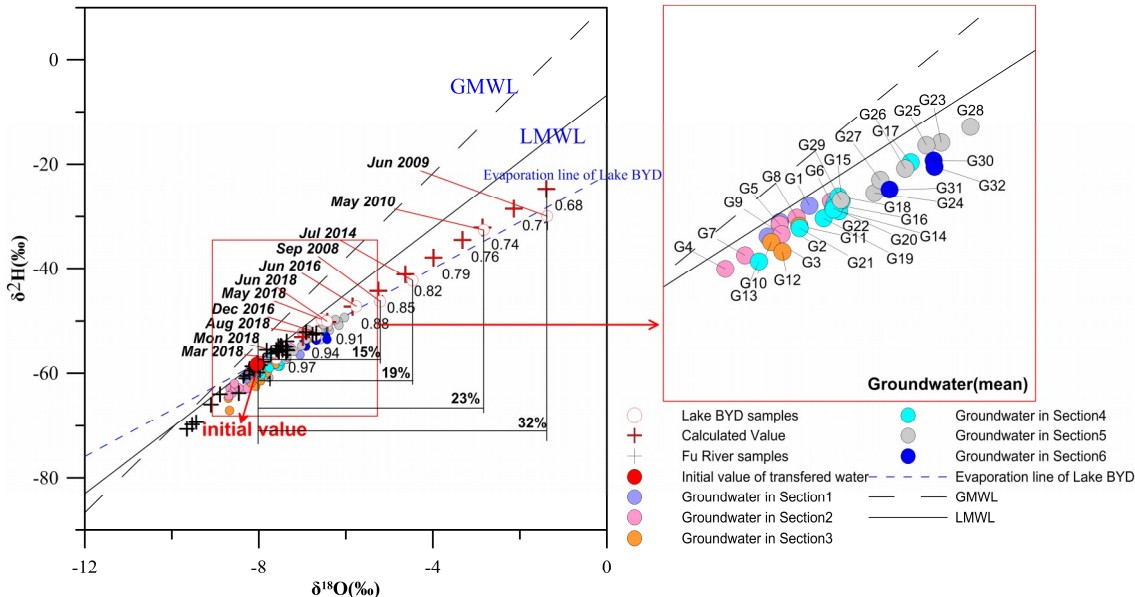

**Figure 8.** The relationship between $\delta^2$H and $\delta^{18}$O in surface water and groundwater, and the evaporation model of the lake water. The global meteoric water line (GMWL) is $\delta^2$H = 8.13$\delta^{18}$O + 10.8 [22]. The local meteoric water line (LMWL) is $\delta^2$H = 6.356$\delta^{18}$O − 6.769 [14]. The evaporation line of the lake water is $\delta^2$H = 4.49$\delta^{18}$O − 22.07 ($R^2$ = 0.98).

### 4.4. Application of End-Member Mixing Analysis

EMMA was used to estimate the contribution from different sources of groundwater. Figure 9 shows $\delta^{18}$O values versus Cl⁻ concentrations for all collected water samples. All groundwater samples are located in the triangle region surrounded by precipitation, Fu River water, the lake water and the lateral groundwater. The mean Cl⁻ concentration in precipitation from 2016 and 2017 observed at an Experimental Station (Luancheng Agro-Ecosystem Experimental Station, CAS) of Shijiazhuang city was used as the precipitation source. The station is located in the piedmont plain area of the NCP, near to the study area. The long-term value of $\delta^{18}$O in precipitation at the Global Network of Isotopes in Precipitation (GNIP) station in Shijiazhuang from 1985 to 2003 was used as the end-member value (data were obtained from http://www.naweb.iaea.org/napc/ih/IHS_resources_gnip.html). The long-term mean values of Cl⁻ concentration and stable isotopes in Lake BYD were referred as values for the other end-member. The Fu River accepted the domestic sewage from Baoding City. The figure (Figure S1) showed that the water chemical ions and stable isotopes were stable in the Fu River during dry seasons (such as January, March, November and December of 2018) when there was no influence of precipitation and transferred water. Therefore, the mean values of Cl⁻ concentration and stable isotopes of Fu River water in January and March of 2018 were taken as values of one end-member of groundwater recharge. The mean values of Cl⁻ concentration and stable isotopes in lateral groundwater upstream of the watershed were included as values of another potential end-member [36]. The four end-members Fu River water (Cl⁻, 372.7 mg/L; $\delta^{18}$O, −8.08‰), the lake water (128.6 mg/L, −5.74‰), lateral groundwater (28.5 mg/L, −8.57‰), precipitation (2.8 mg/L, −7.58‰) were distinct.

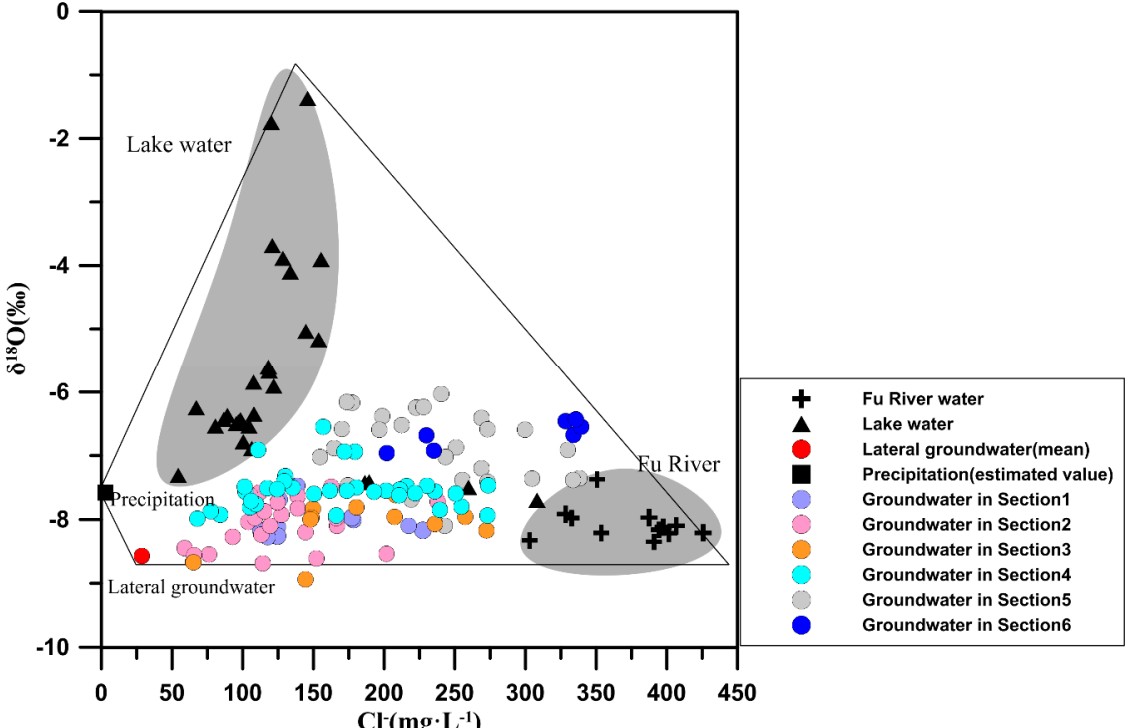

**Figure 9.** $\delta^{18}$O values versus Cl$^-$ concentrations of the groundwater and end-members. The mean values of Cl$^-$ concentration and stable isotopes of the Fu River water in seasons without the influence of the precipitation and transferred water (samples collected in January 2018 and March 2018) were taken as values of one end-member that reflects sewage influence.

Figure 10 shows the $\delta^{18}$O values versus Cl$^-$ concentrations for groundwater collected in different sections. In Sections 1–3, most of groundwater samples plot within the area surrounded by three end-members: lateral groundwater, precipitation and Fu River water. However, the groundwater samples in Section 4 distributed in the triangle composed of precipitation, Fu River water and the lake water, which indicated that the contribution to the lateral groundwater recharge was estimated to be negligible with respect to the artificial water recharge from the Lake BYD in this section. However, most groundwater samples in Sections 5 and 6 deviated from the three end-members triangle, which indicated that the influence of the precipitation became weak. Additionally, a significant number of samples fall outside of the red triangle. Cl$-$ in fertilizer or pesticide might result in the increasing Cl$^-$ concentration in groundwater because the groundwater depth was less than 5 m. In Figure 11, we show that the stable isotopes of groundwater in Sections 5 and 6 fall between the Fu River water and the lake water end-member. This indicated that the mixing of these two sources was dominating the groundwater recharge. Therefore, the stable isotopes of $\delta^2$H and $\delta^{18}$O values were used to estimate the contribution ratios of Fu River water and the lake water (Table 3).

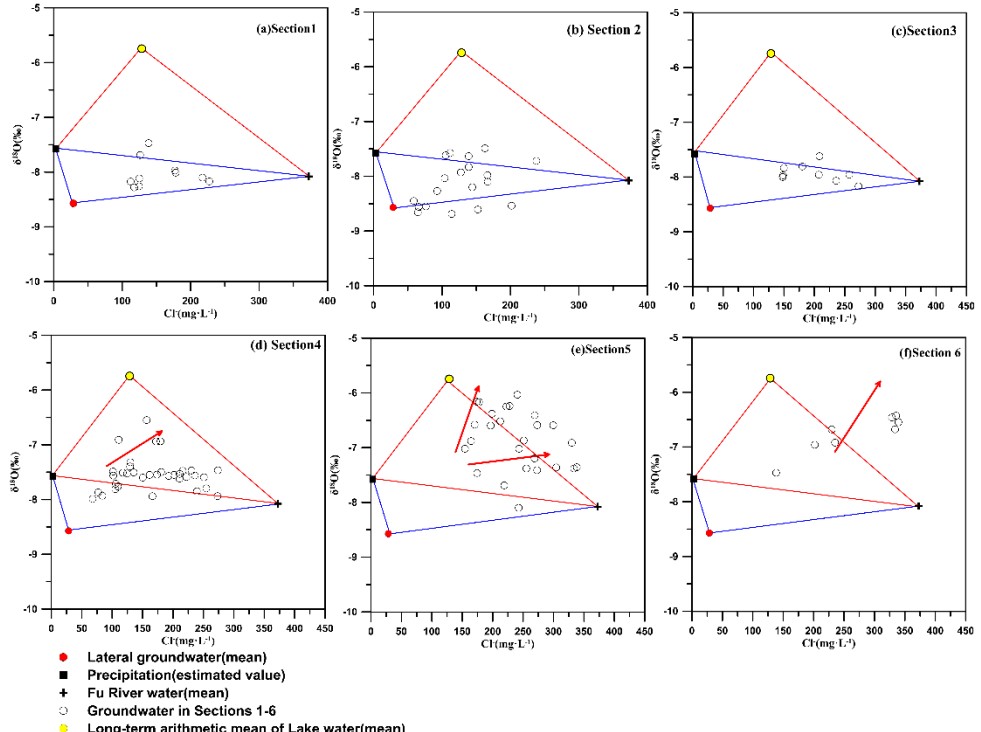

**Figure 10.** $\delta^{18}$O values versus Cl$^-$ concentrations of the groundwater and end-members showing the mixing of different recharge sources. The blue line means the regions influenced by the three end-members of precipitation, lateral groundwater and Fu river water. The red line means the regions influenced by the three end-members of precipitation, lake water and Fu river water.

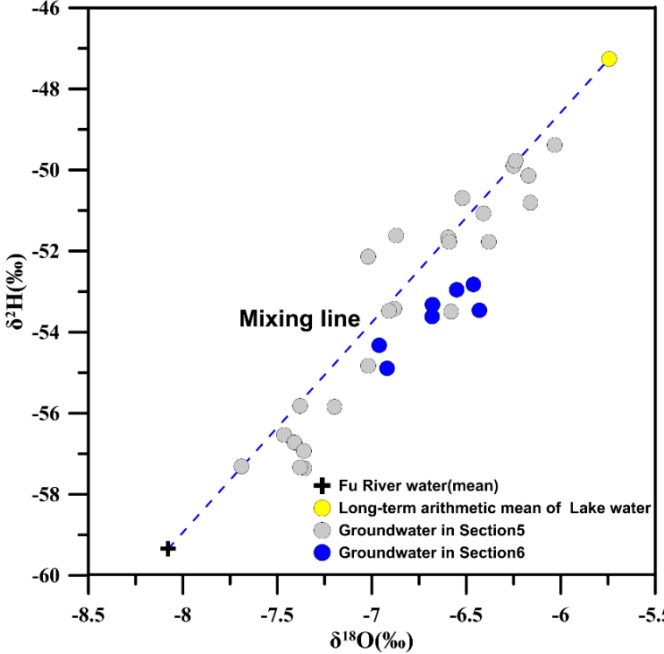

**Figure 11.** Relationship between $\delta^2$H and $\delta^{18}$O of the local groundwater and end-members, showing the mixing in Sections 5 and 6.

**Table 3.** The contribution ratios of recharge sources for the local groundwater in Sections 1–6.

| Section | ID | Distance | PW (%) | LGW (%) | BYD (%) | FR (%) |
|---|---|---|---|---|---|---|
| Sections 1–3 | G1 | 12.4 | 55.2 | 1.1 | / | 43.7 |
| | G2 | −262.1 | 22.5 | 25.2 | / | 52.3 |
| | G3 | −358.1 | 22.7 | 49.1 | / | 28.2 |
| | G4 | −698.1 | 0 | 63.0 | / | 37 |
| | G5 | −320.8 | 28.2 | 31.7 | / | 40 |
| | G6 | 46 | 59.0 | 0.0 | / | 41 |
| | G7 | 357.2 | 7.7 | 77.9 | / | 14.3 |
| | G9 | 619.8 | 52.1 | 16.9 | / | 30.9 |
| | G11 | 44.6 | 47.0 | 12.3 | / | 40.7 |
| | G12 | 493.9 | 17.5 | 15.4 | / | 67.2 |
| Section 4 | G15 | −122.9 | 58.5 | / | 12.4 | 29.1 |
| | G16 | −51.6 | 38.0 | / | 13.7 | 48.2 |
| | G18 | 64.3 | 33.6 | / | 16.4 | 50.0 |
| | G19 | 194.4 | 47.3 | / | 7.4 | 45.3 |
| | G20 | 706.1 | 65.5 | / | 7.4 | 27.1 |
| | G21 | 840.7 | 78.2 | / | 0.0 | 21.8 |
| | G22 | 462.7 | 51.7 | / | 10.5 | 37.8 |
| Sections 5–6 | G23 | −535.5 | / | / | 65.8 | 34.2 |
| | G25 | −533.2 | / | / | 59.7 | 40.3 |
| | G26 | −367.2 | / | / | 51.0 | 49.0 |
| | G27 | −239.7 | / | / | 40.8 | 59.2 |
| | G29 | −51.2 | / | / | 24.6 | 75.4 |
| | G30 | 191.9 | / | / | 62.7 | 37.3 |
| | G31 | 179.9 | / | / | 44.6 | 55.4 |
| | G32 | 476.4 | / | / | 63.2 | 36.8 |

PW: Precipitation; FR: Fu River water; BYD: Lake Baiyangdian water; LGW: Lateral groundwater; Distance: the distance from the Fu River, the negative value represents the north or west bank, and the positive value indicates the south or east bank.

The uncertainty of the concentration of the resultant mixtures and potential end-members caused the uncertainty of the recharge ratios by EMMA. These were evaluated by error propagation analysis [55]. The standard deviations of $\delta^{18}O$ values and $Cl^-$ concentrations for all lateral groundwater samples, Fu River water samples, precipitation, Lake BYD water samples were calculated for their uncertainty. The uncertainty of groundwater was expressed by the standard deviation of the $\delta^{18}O$ and $Cl^-$ concentration of four sampling times at the same sampling site. The estimated contribution ratios and their possible errors are shown in Figure 12. The errors for all groundwater ranged from 2% to 23%. Taken together, these findings suggested that the results of EMMA were reliable.

In Sections 1–3, the errors of contribution ratios for precipitation and lateral groundwater were larger than that for the Fu River water. The varying groundwater depth might be the major reason for the larger uncertainty of contribution ratios. The error of the contribution ratio of Fu River water was relatively small, indicating that the impact of Fu River water on the surrounding groundwater was relatively stable. Even with the errors in the estimated contribution ratios, the general characteristics of the spatial distribution of contribution ratios (Figure 12) did not change significantly due to errors, indicating that the results obtained were fairly insensitive to errors.

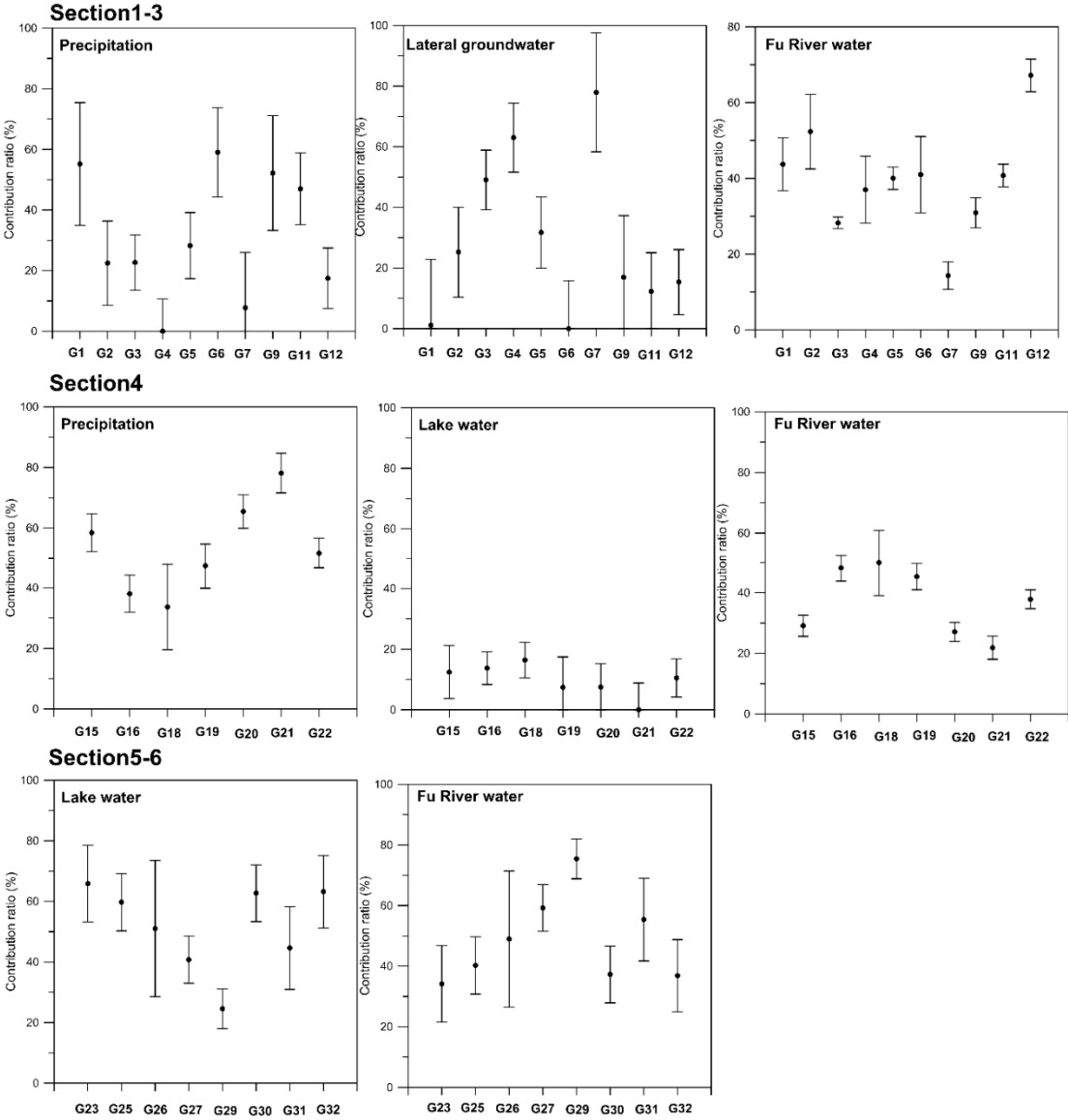

**Figure 12.** The contribution ratios and corresponding estimation errors of recharge sources in all local groundwater sampling sites.

*4.5. Anthropogenic Impact on Surface Water–Groundwater Interaction*

The interaction of surface water and groundwater has been changed greatly by the anthropogenic activities including groundwater over-extraction, waste water discharge and artificial water transfer. Before the 1950s, eight rivers flowed into Lake BYD and groundwater also discharged into the lake. With the development of agriculture, river water was stored in reservoirs in mountain areas and groundwater in the plain area was over-exploited to cover the increasing water need since the 1960s [35,59]. As there is almost no natural water flow in the river, waste water in rivers and transferred water in the lake became the major water resources recharging the local groundwater. The levels of the river and the local groundwater, water chemistry and stable isotopes testified the impact of surface water (Figures 5–8). The over-exploitation induced increasing contribution ratios of lateral groundwater and Fu River water. This was particularly evident for Sections 1–3, where the groundwater table was deep and the contribution of Fu River water ranged from 14% to 67% (Table 3). The contribution of lateral groundwater recharge was estimated negligible further downstream due to the influence of Lake BYD supported by water transfer. The influence of Lake BYD on groundwater extended to

Section 4, which is about 16 km upstream of the lake, with a contribution ratio ranging from 7% to 16% to the total water recharge. Precipitation and Fu River water were the main recharge sources, the contribution ratios ranged from 34% to 78% and 22% to 50%, respectively. In regions near the lake, the contribution ratios of the lake water ranged from 25% to 66% and the contribution ratios of the Fu River water ranged from 34% to 75%. This indicated that Lake BYD and the Fu River formed a direct recharge source of the surrounding groundwater.

The perennial waste water discharge river has an impact on groundwater of both sides of the river. Figure 13 shows the relationship between the contribution ratio of the Fu River water as a function of the distance from the river. There is no clear correlation for the contribution ratio of the river with the distance in Sections 1–3, suggesting the disturbance of groundwater over-exploitation and lateral groundwater flow on the river–groundwater system. However, in Sections 4–6, the contribution ratio of river seepage decreased with increasing distance from the river to both sides. The river recharging the groundwater extends as far as 1400 m to the south and 400 m to the north in Section 4. The river had an influence range of about 900 m on both sides in Sections 5 and 6.

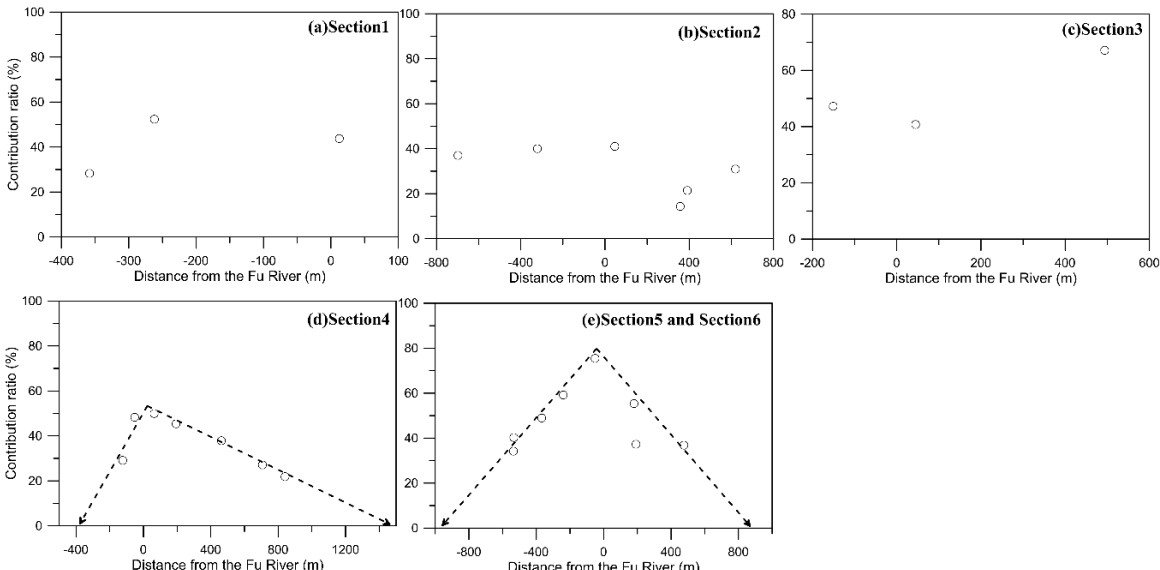

**Figure 13.** Relationship of river water contribution ratios and distances from Fu River to groundwater sampling sites. Distance from the Fu River: the negative value represents the north or west bank, and the positive value indicates the south or east bank.

## 5. Conclusions

We used hydrochemistry and environmental isotope ($\delta^2$H, $\delta^{18}$O) tracers to study the interaction of surface water and groundwater in a river–lake–groundwater continuum system affected by groundwater over-extraction, waste water discharge and water transfer in the Lake BYD area of the North China Plain. Under the influence of groundwater over-extraction, the groundwater table declined, and surface water (freshwater and waste water) in the river–lake–groundwater continuum system became the main recharge source of local groundwater.

We used the end-member mixing analysis method to estimate the contribution ratios of different recharge sources to the groundwater. Seepage from the waste water of the Fu River contributed a significant amount of water to the underlying aquifer along its channel, with the contribution ratio reaching as high as 75%. The region affected by groundwater over-exploitation and artificial water transfer were classified. In the upper reach of the river, groundwater over-exploitation induced an increase of contribution ratios of lateral groundwater, particularly in the region with relatively large groundwater depth. The contribution ratio of lateral groundwater ranged from 12% to 78% with a mean value of 30%. In the middle reach, with a 16 km distance from Lake BYD, the impact

of lateral groundwater was estimated to be negligible due to artificial water transfer to the Lake BYD. The average contribution ratio of the lake water to groundwater near the lake was 52%, with a contribution ranging from 25% to 66%. Thus, Lake BYD supported by water transfer forms a direct recharge source of the surrounding groundwater.

The findings of this study will help to improve understanding of the interaction of surface water and groundwater affected by anthropogenic activities. This study is of great significance for water resources management and groundwater protection under intensified anthropogenic influences.

**Supplementary Materials:** The following are available online at http://www.mdpi.com/2073-4441/11/3/539/s1, Table S1: Statistics of water transfer for Lake BYD from 1981 to 2018. Figure S1: The boxplots of $Cl^-$ (a), $\delta^{18}O$ (b) and $\delta^2H$ (c) for all Fu River water samples collected from January 2018 to January 2019.

**Author Contributions:** Conceptualization, S.W., R.Y. and X.S.; Data curation, M.Z.; Formal analysis, M.Z. and S.W.; Investigation, M.Z., X.K., W.F. and X.Z.; Methodology, M.Z., S.W., W.F., R.Y. and X.S.; Project administration, S.W.; Supervision, S.W.; Validation, M.Z.; Writing—original draft, M.Z.; Writing—review and editing, S.W., X.K., W.Z., X.Z., M.S. and R.Y.

**Funding:** This research was funded by the National Key R&D Program of China (2018YFC0406502; 2018YFD0800306; 2016YFD0800100), the Key R&D Program of Hebei (18273604D), the Program of National Natural Science Foundation of China (41471028), the 100-Talent Project of Chinese Academy of Sciences.

**Acknowledgments:** We would like to acknowledge the assistance from the students and staff of the Center for Agricultural Resources Research, IGDB, CAS. We would like to extend special thanks to the editor and the anonymous reviewers for their valuable comments in greatly improving the quality of this paper.

**Conflicts of Interest:** The authors declare no conflict of interest.

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
