# Peer review of "Interaction of Surface Water and Groundwater Influenced by Groundwater Over-Extraction, Waste Water Discharge and Water Transfer in Xiong’an New Area, China"

_water, doi:10.3390/w11030539_

Round 1

Reviewer 1 Report

The paper discusses the application of field data and water sample analysis to better understand the interaction between the surface and groundwater at the river-lake groundwater system in the Baiyangdian Lake catchment. I think the paper is well structured and written. However, there are unclear sentences that need improvement and there is a need for some clarifications which I list below:

·         I picked errors in the following sentences, but I suggest authors review the text thoroughly:

o   Abstract: the sentence “In this paper, a building new district in the North of China”

o   Introduction (Paragraph 3): “Chloride CL- has strong hydrophilicity and conservative chemical properties, which can be…” replace can be by “were”

o   Introduction (Paragraph 4, Line 3): “have let to dry”, change to “drying”

o   Introduction (Paragraph 4, Line 5): “the lake water was recharge”, change to recharged

o   Introduction (Paragraph 4, Line 19): “groundwater and it influencing”, change it to its

o   Change title of Section 2.3 to “Hydrological setting”

o   Correct title of Section 4.1: “Stageand”

o   Figure 5: Correct the label of the horizontal axis  

o    Section 4.2, Paragraph 2, Line 2: spatial rather than special

o   Section 4.4, Paragraph 1, Lines 6 and 7: not clear

o   Acknowledgment: change “stuffs” to “Staffs”

·         The followings need clarification or correction:

o   Introduction, Line 13: Unless the groundwater is in the soil zone, its decline won’t affect the runoff.

o   For international readers, I suggest to include the Yellow River in the Figure. If not possible mention its location with reference to the study area in the text.

o   Section 3.1 cross references Figure 1b. I think this is incorrect.

o   Section 3.2, Paragraph 1, Line 7: the parameter (f) is the remaining fraction of the reservoir or the materials?

o   Section 4.1, Paragraph 1. The definition of the different river sections should be clearer, i.e. what are the spatial extents of the different sections? Also on Line 7, it is mentioned that at the outlet of Fu River, the river water is higher than the main water table level. This is not consistent with Figure 5f (Section 6). Finally the last sentence of this paragraph mentions that “the groundwater table depth is larger than 10 m in Section 1 to 3 but less than 5 m in Section 4 to 6”. These values are consistent with Figure 6a but not with Figure 5

o   Figure 5: Correct the label of the horizontal axis  

o   Section 4.3: it mentions that the sequence of δ values were found in decline as BYD Lake > groundwater> Fu River. This is not consistent with the mean values in Table 2, which is not in agreement with the discussion in the paragraph that follows Table 2. Please clarify. Also please add the full description of the acronym “CV” (Coefficient of variation) when it is first mentioned in the text.

Author Response

Dear reviewer,

Thank you very much for your consideration. We have revised our manuscript according to your comments and suggestions. We have point-by-point response to your comments in the upload Word file. Best wishes!

Kind regards,

Meijia Zhu

Reviewer 2 Report

The manuscript presents in a comprehensible form a consistent set of geochemical data as cation and anion composition of waters as well as stable isotope composition in order to quantify groundwater over-extraction in a high populated semi-arid region and limited resources of water belonging to the Baiyangdian Lake, Fu Rive, Baoding City and Anxin County (North China Plain).

I recommed publication of the manuscript after changes are done, as indicated in the attached PDF file with annotations.

Author Response

(The authors gave the same response as above.)

Reviewer 3 Report

Manuscript must be rewritten clearly. Many sentences are confused. Details on computation method and chemical quality of groundwater and surface water  are missing. What type of contaminant the authors are dealinling with? Organic compouds? pesticides? halogen compouds? ammonia? Why these compous did not affect water EC?
I suggests to replace sentences such as "Understanding impacts of intensified anthropogenic activities.." with " To understand the interaction between the surface water and groundwater ... " 

Author Response

(The authors gave the same response as above.)

Round 2

Reviewer 1 Report

The revised manuscript addressed the points I raised. However, the text added need English improvement. I suggest these to be revisited before the full acceptance of the manuscript.

Author Response

(The authors gave the same response as above.)

Reviewer 2 Report

I recommed to accept the manuscript in present form. Please check once more the english spelling.

Author Response

(The authors gave the same response as above.)

Reviewer 3 Report

Authors have developed an interesting topic supported by field measurements and surface water/groundwater monitoring. Anyway the manuscript text is very difficult to understand because many sentences do not have a clear meaning. For instance in Abstract, authors wrote:

 “However, the contribution of lateral groundwater recharge disappeared due to the increasing influence of artificial water recharge to the Lake BYD.”

Probably, this sentence should be rewritten as

“However, the contribution to the lateral groundwater recharge was estimated negligible with respect to the natural groundwater recharge from the BYD Lake, which is constantly increasing due to wastewater outflows coming from anthropic activities (i.e., industrial and municipal treatment plants) and irrigation runoffs. “

Moreover in Fig. 5 “River water stage and water table in each section”,

Is "the water level" the river water depth, or is it the height of the water table of groundwater above sea level? What is the meaning of "water stage"?

I suggest an accurate revision of the text by focusing on the meaning of the sentences.

Author Response

(The authors gave the same response as above.)
